# Glutamine Addiction and Therapeutic Strategies in Lung Cancer

**DOI:** 10.3390/ijms20020252

**Published:** 2019-01-10

**Authors:** Karolien Vanhove, Elien Derveaux, Geert-Jan Graulus, Liesbet Mesotten, Michiel Thomeer, Jean-Paul Noben, Wanda Guedens, Peter Adriaensens

**Affiliations:** 1Faculty of Medicine and Life Sciences, Hasselt University, Martelarenlaan 42, B-3500 Hasselt, Belgium; karolien.vanhove@uhasselt.be (K.V.); liesbet.mesotten@zol.be (L.M.); michiel.thomeer@uhasselt.be (M.T.); elien.derveaux@uhasselt.be (E.D.); 2Department of Respiratory Medicine, Algemeen Ziekenhuis Vesalius, Hazelereik 51, B-3700 Tongeren, Belgium; 3Biomolecule Design Group, Institute for Materials Research, Hasselt University, Agoralaan Building D, B-3590 Diepenbeek, Belgium; geertjan.graulus@uhasselt.be (G.-J.G.); wanda.guedens@uhasselt.be (W.G.); 4Department of Nuclear Medicine, Ziekenhuis Oost-Limburg, Schiepse Bos 6, B-3600 Genk, Belgium; 5Department of Respiratory Medicine, Ziekenhuis Oost-Limburg, Schiepse Bos 6, B-3600 Genk, Belgium; 6Biomedical Research Institute, Hasselt University, Agoralaan, B-3590 Diepenbeek, Belgium; jeanpaul.noben@uhasselt.be; 7Applied and Analytical Chemistry, Institute for Materials Research, Hasselt University, Agoralaan Building D, B-3590 Diepenbeek, Belgium

**Keywords:** Lung cancer, metabolism, glutamine, glutaminolysis, pathways, targeted treatment

## Abstract

Lung cancer cells are well-documented to rewire their metabolism and energy production networks to support rapid survival and proliferation. This metabolic reorganization has been recognized as a hallmark of cancer. The increased uptake of glucose and the increased activity of the glycolytic pathway have been extensively described. However, over the past years, increasing evidence has shown that lung cancer cells also require glutamine to fulfill their metabolic needs. As a nitrogen source, glutamine contributes directly (or indirectly upon conversion to glutamate) to many anabolic processes in cancer, such as the biosynthesis of amino acids, nucleobases, and hexosamines. It plays also an important role in the redox homeostasis, and last but not least, upon conversion to α-ketoglutarate, glutamine is an energy and anaplerotic carbon source that replenishes tricarboxylic acid cycle intermediates. The latter is generally indicated as glutaminolysis. In this review, we explore the role of glutamine metabolism in lung cancer. Because lung cancer is the leading cause of cancer death with limited curative treatment options, we focus on the potential therapeutic approaches targeting the glutamine metabolism in cancer.

## 1. Introduction

Cancer cells exhibit metabolic alterations that distinguish them from healthy cells, which are recognized as one of the 10 hallmarks of cancer [1]. An altered metabolism helps cancer cells to sustain high proliferative rates, even in a hostile environment characterized by hypoxia and insufficient nutrient supply, due to poor vascularization [2]. Metabolomics, i.e., the study of small molecule metabolites, has emerged as a powerful tool, as its non-invasive nature and close link to the phenotype makes it an ideal tool in the study of cancer biology, diagnosis, and even in the discrimination between cancer types [3,4,5,6,7,8,9]. 

In the last century, Otto Warburg postulated that even in the presence of oxygen, cancer cells rely on glycolysis, leading to lactate production, via the fermentation of pyruvate [10]. The increased conversion of pyruvate to lactate and the shunt of glycolytic intermediates into biosynthetic pathways limits the availability of pyruvate to form acetyl-CoA in the mitochondria that is needed to maintain the tricarboxylic (TCA) cycle [11,12,13]. In addition, TCA cycle intermediates are diverted in anabolic processes that are operational in growing and proliferating cells [14]. To compensate for the reduced influx of glucose-derived carbon towards acetyl-CoA and the permanent drain of TCA cycle intermediates into biosynthetic pathways, proliferating cancer cells increase their uptake of glutamine, a nonessential amino acid [15,16].

In marked contrast to normal cells, lung cancer cells become addicted to glutamine as a consequence of aberrantly activated oncogenes, and the loss of tumor suppressors [17]. These alterations do not only result in the import of glutamine via the upregulation of glutamine transporters, but they also promote the expression of metabolic enzymes involved in the metabolism of glutamine [18]. For example, a commonly deregulated pathway in lung cancer is the extension of functions of MYC by translocations and amplifications [19]. The c-MYC oncogene induces the transcription of glutamine transporters, ASCT2 and LAT1, and the expression of glutamine-utilizing enzymes such as glutaminase (GLS1) [18,20]. Following the transport of glutamine into the cell, the first step of glutaminolysis, i.e., the catabolism of glutamine, is the GLS1-mediated conversion of glutamine into glutamate. Glutamate is subsequently converted to α-ketoglutarate (α-KG) by either glutamate dehydrogenase (GLUD) or aminotransferases.

Glutamine and its metabolic conversion products glutamate and α-KG all contribute as nitrogen and/or carbon sources to the biosynthesis of important cellular constituents such hexosamines, purines, pyrimidines, amino acids, fatty acids, and glutathione [21,22,23]. Furthermore, the efflux of glutamine, coupled to the import of essential amino acids such as leucine, results in the activation of the mammalian target of rapamycin (mTOR) pathway, which promotes the synthesis of proteins and inhibits autophagy [17].

## 2. Glutamine Metabolism in Lung Cancer

The uptake of circulating glutamine into proliferating and cancer cells is mediated by the alanine-serine-cysteine-transporter-2 (ASCT2 or SLC1A5) [24]. Hassanein et al. demonstrated the overexpression of this receptor in squamous cell carcinoma, adenocarcinoma, and neuroendocrine lung tumors [25]. ASCT2 was identified as the primary importer of glutamine, and its inhibition resulted in reduced growth and proliferation [25].

The imported glutamine can be used or can be exchanged through the l-type amino acid transporter (LAT1 or SLC7A5) for essential amino acids such as (iso)leucine, valine, methionine, tryptophan, and phenylalanine [26]. LAT1 expression in non-small cell lung cancer has been demonstrated by Takeuchi et al., and inhibition of LAT1 by 2-aminobicyclo-(2,2,1)-heptane-2-carboxylic acid (BCH) resulted in reduced lung cancer cell viability [27,28].

The glutamine flux, which is a balance between the uptake of glutamine by ASCT2 and its subsequent export by LAT1, results in a high intracellular availability of essential amino acids. Once in the cell, glutamine is a substrate of several glutamate-producing enzymes, such as mitochondrial glutaminase and cytosolic enzymes that are involved in the biosynthesis of nitrogenous metabolites. Glutamine-derived glutamate may undergo transport back out of the cell in exchange for cystine by the xCT (SLC7A11) antiporter [29]. Using a labeled glutamate-analogue, the presence of xCT has been used to detect lung cancer in an exploratory trial by Baek et al. [30]. The role of the receptor has been further explored by experimental targeting of the xCT receptor with sulfasalazine, which resulted in a decreased cell proliferation in several lung cancer cell lines, and in mice [29].

### 2.1. Glutaminolysis

In contrast with the initial hypothesis of Warburg, functional mitochondria are present in cancer cells, and they act as biosynthetic factories [14]. An important metabolic pathway that occurs in the matrix of the mitochondrion is the TCA or Krebs cycle. The TCA cycle is composed of eight biochemical reactions that: (i) oxidize fuel sources to provide ATP; (ii) support the synthesis of macromolecules such as lipids and; iii) regulate cellular redox balance. In addition, the TCA cycle provides precursors of certain amino acids. Continued functioning of the TCA requires the replenishment of intermediates (anaplerosis) that are diverted for synthesis of macromolecules and ATP. Anaplerosis is accomplished via two major pathways: the glutaminolysis and carboxylation of pyruvate to oxaloacetate (OAA) via pyruvate carboxylase (PC). In mammals, two major pathways exist for the metabolic conversion of glutamine to α-ketoglutarate (1) the glutaminase II pathway, in which glutamine is transaminated to α-ketoglutarate, followed by its hydrolysis to α-ketoglutarate and free ammonia (NH_4_^+^) catalyzed by ω-amidase, and (2) the hydrolysis of glutamine to glutamate by glutaminases (GLS) located in the mitochondria, followed by the conversion of glutamate to α-ketoglutarate and NH_4_^+^, by glutamate dehydrogenase (GLUD) or glutamate-linked aminotransferases such as alanine aminotransferase (ALT2) and aspartate aminotransferase (AST2) [31,32].

In mammals, two different isoforms of GLS are expressed, i.e., kidney-type (GLS1) and liver-type (GLS2). Upregulation of the GAC isoform, a shorter splice variant of GLS1, is common in lung cancer [33]. In contrast, the role of GLS2 is still not completely understood, and it seems to be context-dependent [34]. As mentioned before, the action of GLS and the oxidative deamination of glutamate results in the release of ammonia [35]. NH_4_^+^, a potential toxic molecule, is cleared by diffusion or transport out of the cell, or through its incorporation in the α–ketoacids, oxaloacetate and pyruvate, to generate aspartate and alanine, respectively [22]. NH_4_^+^ is a diffusible regulator of autophagy, which supports cancer cell survival during metabolic stress by the degradation of intracellular proteins and organelles of cancer and stromal cells, effectively recycling macromolecules into metabolic precursors that support processes that are essential for cell survival [36,37]. Indeed, emerging evidence indicates that autophagy in stromal cells such as endothelial cells and cancer-associated fibroblasts provides cancer cells with an abundant source of glutamine [38]. A difference in glutamine dependency for K-ras driven lung cancer cells when studied in vitro and in vivo was demonstrated by Davidson et al., supporting the evidence of the complex interaction between the tumor and the stromal cells on the process of tumor proliferation [39]. More specifically, when tissue glutamine levels are low, certain types of stromal cells such as cancer-associated fibroblasts and immune cells such as tumor-associated macrophages surrounding the tumor cells might play a role in a higher production of glutamine compared to normal fibroblasts [40].

NH_4_^+^ can also be incorporated into carbamoyl phosphate by mitochondrial carbamoyl phosphate synthetase 1 (CPS-1) [23]. Surprisingly, lung cancer cell death induced by CPS-1 silencing resulted merely from pyrimidine depletion, rather than from cytotoxic ammonium accumulation [41]. It seems that mitochondrial CPS-1 enables an unconventional flow of nitrogen from the mitochondrial matrix (where carbamoyl phosphate is produced from the ions ammonium and bicarbonate) to the cytosol (were carbamoyl phosphate is used in pyrimidine synthesis). By doing so, the rate-limiting step in pyrimidine synthesis catalyzed by CPS-2 is omitted, and glutamine as the ‘natural’ nitrogen donor in pyrimidine synthesis is spared for glutaminolysis.

Glutamine-derived α-KG provides carbons towards the biosynthesis of diverse biosynthetic precursors in the TCA cycle. Just as glycolytic intermediates are harnessed to support anabolic pathways, lung cancer cells rewire their TCA cycles to prioritize cell growth over ATP generation by OXPHOS. Indeed, the highly upregulated GLS1 isoform in non-small cell lung cancer (NSCLC) finally results in increased levels of TCA cycle intermediates [33,42]. In contrast, the suppression of GLS1 by small hairpin RNA (shRNA) results in reduced growth rate in lung cancer cells [43].

In general, the fate of α-KG is determined by environmental factors, such as hypoxia and nutrient deprivation (Figure 1) [44,45]. In normoxic conditions and an adequate glucose supply, the mitochondria of cancer cells can shunt glutaminolytic α-KG into the oxidative phosphorylation (OXPHOS) to generate ATP, or into the production of oxaloacetate (OAA) via oxidative reactions in the TCA cycle [46]. The condensation of glutamine-derived OAA and glucose-derived acetyl-CoA results in the production of citrate and other intermediates of the TCA cycle [47]. Citrate is exported to the cytosol and cleaved by ATP–citrate lyase into acetyl-CoA to be used for lipogenesis and OAA. The cytosolic OAA is reduced into malate by NADH-dependent malate dehydrogenase (MDH1). Malate returns to the mitochondrion, or it undergoes oxidation by cytosolic malic enzyme 1 (ME1), resulting in the production of pyruvate, CO_2_ and reduced nicotinamide adenine dinucleotide phosphate (NADPH). Along with the NADPH-producing reactions in the pentose phosphate pathway, this reaction provides the reducing power for lipid biosynthesis [47]. Pyruvate in the cytosol may undergo a reduction into lactate by lactate dehydrogenase, further increasing the acidity of the extracellular environment. In analogy with glycolysis, the complete metabolism of glutamine into the end product, lactate, has been termed glutaminolysis. Analogous to citrate, malate can also leave the TCA cycle to fulfill biosynthetic needs, e.g., by being converted into pyruvate and NADPH by ME1, as previously mentioned. Mitochondrial OAA can only leave the TCA cycle after its conversion into aspartate by mitochondrial AST2. In the cytosol, aspartate supports the synthesis of nucleotides, or it contributes to the generation of asparagine [21].

Alternatively, under conditions of hypoxia or glucose starvation, glutamine-derived α-KG undergoes a reductive carboxylation by NADPH-dependent isocitrate dehydrogenase 2 (IDH2) to generate citrate for lipid synthesis [44,48]. The mitochondrial NADPH/NADP^+^ ratio required to fuel the reductive reaction through IDH2 can arise from the increased NADH/NAD^+^ ratio existing in the mitochondria during hypoxia, with the transfer of electrons from NADH to NADP^+^ to generate NADPH through mitochondrial transhydrogenase [49].

In contrast with cytosolic IDH1, overexpression of IDH2 has not been described in patients with lung cancer [48]. IDH1 plays a role in the antioxidant system by producing NADPH. Upregulation of IDH1 may be an adaptive alteration for lung cancer cells to antagonize and survive oxidative stress. On the other hand, Metallo et al. demonstrated that IDH1 can convert cytosolic α-KG to isocitrate, resulting in the consumption of NADPH. Furthermore, using labelled glutamine tracers under hypoxia, they detected increased amounts of the isotopic label in TCA cycle intermediates such as citrate, aspartate, malate and cytosolic acetyl-CoA, thereby providing the evidence that cytosolic isocitrate enters the TCA cycle [50]. In addition to glucose deprivation and hypoxia, Corbet et al. documented that proliferating cancer cells, chronically exposed to acidic conditions, increased the glutamine metabolism to sustain lipogenesis through IDH1 dependent reductive carboxylation of α-ketoglutarate concomitant with fatty acid catabolism [51,52].

### 2.2. Glutamine as a Nitrogen Source in the Synthesis of Purines and Pyrimidines

Glutamine is an indispensable donor of nitrogen for the synthesis of nucleobases. Two molecules of glutamine are consumed in the biosynthesis of the parent nucleotide inosine monophosphate (IMP) which is the precursor of adenosine monophosphate (AMP) and guanosine monophosphate (GMP). The synthesis of GMP requires a third nitrogen molecule of glutamine. The first reaction in the synthesis of pyrimidines is the condensation of glutamine-derived nitrogen, ATP and bicarbonate resulting in the production of carbamoyl phosphate. The synthesis of nucleotides from exogenous glutamine has been observed in human lung cancer samples cultured ex vivo [43]. A complete description of the nucleobases biosynthesis is out of the scope of this review but has been described in detail by Harvey and Ferrier [53].

### 2.3. Glutamine is Used to Generate UDP-GlcNAc

Lung cancer cells display a wide range of glycosylation alterations compared with their normal counterparts [54]. In the hexosamine biosynthetic pathway (HBP), both glucose and glutamine are essential for the production of uridine diphosphate-N-acetylglucosamine (UDP-GlcNAc), a metabolite that serves as an essential building block for the synthesis of glycoconjugates such as glycosaminoglycans, glycolipids, and glycoproteins. Through fructose-6-phosphate, glutamine, acetyl-CoA and uridine, the HBP is well-positioned to sense the four macromolecules of life (carbohydrates, amino acids, lipids, and nucleotides) [55]. UDP-GlcNAc is essential for both N- and O-GlcNAcylation. Of particular interest in lung cancer is O-GlcNAcylation which refers to the enzymatic addition of the N-acetylglucosamine moiety of UDP-GlcNAc to the hydroxyl group of serine and threonine residues, which may also be targets for phosphorylation. The O-GlcNAcylation of cellular proteins, oncogenes, and tumor suppressor genes, can significantly impact tumor growth, proliferation, invasion, metastasis, and metabolism. O-GlcNAcylation is dynamically regulated by O-GlcNAcylation transferase (OGT), which is responsible for O-GlcNAc addition, and O-GlcNAcase, which is responsible for O-GlcNAc removal. Mi et al. demonstrated an increased expression of OGT and an elevated O-GlcNAcylation in lung cancer tissue. In contrast, OGA levels were not significantly different between cancer tissue and adjacent normal tissue [56].

In lung cancer cells C-MYC is frequently expressed at constitutive high levels, which results in glutamine addiction and the expression of almost all glycolytic enzymes [20,57]. After the activation of C-MYC by extracellular tyrosine kinase, the degradation of c-MYC is controlled by phosphorylation of specific Ser62 and Thr58 sites. Increased O-GlcNAcylation of the threonine site can compete with phosphorylation, resulting in an impaired degradation of c-MYC, and thus stabilization that results in the constitutive transcription of many genes and transporters that are involved in cancer metabolism [58]. A complete description of how O-GlcNAcylation stabilizes oncogenic signaling in cancer is out of the scope of this review, but it has been extensively described by Jozwiak et al. [58].

### 2.4. Glutamine as a Precursor of Non-Essential Amino Acids

After the deamidation of glutamine into glutamate by glutaminase (GLS), its remaining nitrogen can be transferred to different α-ketoacids by a family of cytosolic aminotransferases. The transfer of the amino nitrogen by ALT1, AST1, and phosphoserine aminotransferase (PSAT1) results in the production of alanine, aspartate, and phosphoserine, respectively. Phosphoserine is a precursor for the biosynthesis of glycine and cysteine, which are both needed in the synthesis of glutathione. Furthermore, glycine is a source of specific carbon and nitrogen atoms in the purine ring. Aspartate is a precursor for the synthesis of asparagine, and a substrate in the synthesis of purines and pyrimidines, as well as a protein building block.

### 2.5. Glutamine in Redox Control

Glutamine-derived glutamate is a component of the main anti-oxidation factor, glutathione, and it is also responsible for intracellular cystine levels, as glutamate export by the xCT antiporter is coupled to cystine import [29]. Once in the cell, cystine undergoes a reduction into cysteine, an amino acid that is also needed in the synthesis of glutathione [21,29].

High concentrations of cysteine have been demonstrated by Krepela et al. in lung cancer tissue [59]. These high levels have been confirmed by Fahrmann et al. [60]. Furthermore, these authors described higher levels of the cysteine generating enzyme cystathionine gamma-lyase. After the condensation of homocysteine and serine, the resulting cystathionine is cleaved by cystathionine gamma-lyase, resulting in an additional source of cysteine. Reduced glutathione (GSH), a tripeptide of glutamate–glycine and cysteine, is synthesized in the cytosol of cancer patients, and a small percentage is imported into the mitochondria, where it functions as a scavenger for reactive oxygen species (ROS) that are generated during mitochondrial oxidative metabolism [61]. High levels of ROS initiate cell death signaling pathways and result in production of hydroxyl radicals, which can directly damage DNA, proteins, and lipids. To prevent toxic levels of ROS, tumor cells increase their antioxidant capacity. GSH neutralizes hydrogen peroxide with support of the glutathione peroxidase enzyme. In turn, oxidized glutathione (GSSG) is reduced by NADPH and glutathione reductase to regenerate GSH. Metabolic profiling of lung cancer tissue by Rocha et al. and several other studies that were summarized in a review by Gamcsik et al. showed higher glutathione levels in lung cancer tissue, as compared to disease-free lung tissue [6,61].

### 2.6. Glutamine in Cell Signaling and Metastasis

The metabolism of glutamine results in the stimulation of several signaling pathways that promote cell growth and proliferation. Glutamine is required for the activation of the mammalian target of rapamycine (mTOR), a key signaling node that regulates protein translation, cell growth, and autophagy [62]. In the absence of amino acids, mTOR is unresponsive to growth factor stimulation. The uptake of glutamine through ASCT2, followed by its rapid efflux through LAT1 in exchange for essential amino acids (EAA) such as leucine is the rate-limiting step for mTOR activation in cancer cells [62]. In hepatoma cells, the silencing of ASCT2 inhibits signaling by mTOR to the translational machinery, and abrogates growth and survival [63]. Recently, it was demonstrated that the generation of α-KG, resulting from the catabolism of glutamine, was critical for the activation of mTOR in cancer cell lines of cervical carcinoma and osteosarcoma [64]. Targeting glutamine uptake and glutaminolysis in cancer patients has the potential to suppress mTOR signaling, even in the presence of aberrant growth factor stimulation. Targeting mTOR signaling may have added value in the treatment of lung cancer, as mutations that promote mTOR signaling, such as alterations in the PI3K-AKT pathway, are among the most frequent activations in lung cancer [65,66]. Another major pathway that is involved in lung cancer is the mitogen-activated protein kinase (MAPK, i.e., RAS-RAF-MEK-ERK) pathway [67]. In contrast with the PI3K-AKT-mTOR cascade, studies on the regulation of cell proliferation by glutamine in the MAPK pathway are controversial [62]. Indeed, Traves et al. demonstrated that inhibition of the MEK/ERK pathway did not interfere with the metabolism of glutamine in activated macrophages [68]. However, Kim et al. revealed that glutamine promotes growth, migration, and differentiation in human dental pulp cells, as a result of an activated MAPK pathway [69]. In contrast with these findings, Yuan et al. revealed that glutamine increased the activities of GLS and GLUD by modulating the mTOR/S6 and MAPK pathways in ovarian cancer cells [70]. However, the role of inhibition of glutamine-induced signaling in lung cancer has not been described. A different route through which glutamine influences cellular signaling is the previously described HBP. In this pathway, both glucose and glutamine are essential for the production of UDP-GlcNAc, a metabolite that serves as a substrate for N-linked or O-linked protein glycosylation. The glycosylation of oncogenes, tumor suppressor genes, and other proteins such as interleukins involved in signaling pathways may significantly impact tumor growth, cell proliferation, angiogenesis invasion, and metastasis [58,71]. Due to the fact that glutamine is an essential amino acid for proliferating tumor cells, its role in cancer progression has been extensively investigated. The silencing of GLS1 was able to counteract the induction of the epithelial-to-mesenchymal transition (EMT), mediated by growth factors such as tumor growth factor β (TGF-β). The EMT process has been extensively reported to contribute to cancer progression and metastasis, and importantly, GLS1 silencing impaired growth and metastasis formation [72]. Hereby, the role of glutamine metabolism is of importance, by contributing to the functional regulation of the tumor suppressor gene p53 [73]. Inhibition of glutamine metabolism, either by GLS1 silencing or glutamine deprivation, promotes p53 activation [72]. In lung carcinoma, a loss of p53 functions is found to promote EMT, specifically by decreasing the levels of microRNA (miRNA)-34 [74]. Thus, targeting GLS could be an effective strategy to block the EMT process, and therefore impair the invasive abilities of aggressive lung cancer cells. Furthermore, TGF-β signaling is reported to play a crucial role in myofibroblast proliferation. Upregulation of the TGF-β pathway seems to contribute to an interaction between myofibroblasts and lung cancer cells, leading to increased tumor progression [75]. This link is supported by research of Shi et al. who showed that inhibition of TGF-β signaling results in a reduction of cell proliferation and tumor growth in lung cancer cells [76]. The critical role of glutamine in this signaling pathway is recently documented: TGF-β induces glutaminolysis by stimulating GLS1 expression, and it hereby controls myofibroblast differentiation [77]. In addition to its well-described role in supporting tumor growth by providing metabolites, the metabolism of glutamine may also contribute to tumor metastasis and progression [72]. The TGF-β pathway seems not to only interact with fibroblasts, but also with tumor-associated macrophages that rely on this pathway to stimulate tumor metastasis in lung cancer [78].

## 3. Therapeutic Perspectives of Glutamine Metabolism

The importance of glutamine for critical processes in malignant cells makes this branch of metabolism an attractive target for therapeutic strategies (Table 1 and Figure 2). Due to the diverse roles of glutamine, the consequences of targeting glutamine metabolism depends on the exact process being impacted. Lowering the plasma concentration of glutamine, inhibition of glutamine transport, the development of glutamine-mimetics, and the inhibition of critical enzymes such as GLS and GLUD are interesting pharmacological strategies to inhibit the glutamine metabolism in lung cancer cells. It is well known that the glutamine metabolism contributes not only to cancer cell proliferation, but also leads to the development of adaptation to oxidative stress by the production of glutathione. Modulation of the redox activity by targeting glutathione production influences the concentration of ROS that affects gene transcription, resulting in cell growth and apoptosis signals [79]. As a consequence, inhibition of the glutamine pathway results in lower levels of glutathione, and an extensive failure to combat oxidative stress. Therefore, the inhibition of glutamine metabolism may be a promising adjuvant strategy for suppressing the development of resistance to conventional cancer treatments. As an example, dual inhibition with the GLS-inhibitor bis-2-(5-phenylacetamido-1,3,4-thiadizol-2-yl)ethyl sulfide (BPTES) and its pyrimidine analogue elicited cell death synergistically through cell cycle arrest, and resulted in a remarkable anticancer effect in a preclinical NSCLC model [42].

### 3.1. Depletion of Plasma Glutamine

Glutamine is the most abundant amino acid in plasma, and the depletion of the glutamine supply by lowering the plasma glutamine concentration may be a therapeutic option. Infusion of glutaminase into the bloodstream successfully reduced glutamine levels in large animal experiments, but also resulted in fatal gastro-intestinal side-effects [80]. ʟ-asparaginase has been demonstrated to lower plasma glutamine concentrations in acute lymphoblastic leukemia by removal of the amide nitrogen from glutamine to form glutamate [81]. The extent to which plasma glutamine depletion contributes to improvement of outcome in lung cancer patients remains an open question. A major drawback of strategies that reduce the concentration of glutamine is the potential development of immunosuppression, as glutamine is known to be essential for the proliferation of lymphocytes, macrophages, and neutrophils [88].

### 3.2. Glutamine Transport Inhibitors

As previously mentioned, the overexpression of glutamine transporters accounts for the increased uptake of glutamine by lung cancer cells. The inhibition of glutamine transport is another strategy to for restricting glutamine metabolism. Pharmacological targeting of the ASCT2 receptor in lung cancer patients by γ-l-glutamyl-p-nitroanilide (GPNA) only led to glutamine starvation in cells with ASCT2 overexpression [82]. In this context, the pretreatment evaluation of glutamine uptake by ASCT2, using positron emission tomography with labeled glutamine-analogues, may select patients that benefit from GPNA treatment [89]. In preclinical pharmacological trials, V-9302, another competitive antagonist of the transmembrane glutamine flux, has not only demonstrated attenuated growth and proliferation, but it also increased cell death and oxidative stress which resulted in antitumor responses [83]. Limitation of the glutamine import, not only influences glutamine metabolism, but it also decreases the mTORC1 signaling, resulting in a decrease of lung cancer proliferation. mTORC1 is a signaling node that regulates several processes such as mRNA translation, cell cycle progression, and autophagy. In the absence of glutamine, the exchange of intracellular glutamine, coupled with the import of essential amino acids becomes compromised. As these amino acids are the rate-limiting step for mTORC1 activation, the inhibition of ASCT2 also results in the silencing of this signaling pathway. As a consequence of ASCT2 inhibition, the lower concentration of glutamate also results in the lower concentrations of ammonia, thereby inhibiting the cytoprotective process of autophagy in malignant cells. Another way to inhibit the mTOR pathway activity, by targeting the LAT1 receptor, is the direct inhibition of leucine import by 2-aminobicyclo-(2,2,1)-heptane-2-carboxylic acid (BCH) [28]. More recently, experimental targeting of the xCT (SLC7A11) receptor with sulfasalazine, an approved drug in the treatment of inflammatory bowel and rheumatic diseases, resulted in a decreased cell proliferation in several cell lines, and in mice [29]. These results provide evidence for targeting the cystine–glutamate transporter. Notwithstanding these encouraging results from targeting the glutamine and glutamate transporters in cell lines, a search of the Clinical Trials database did not reveal any ongoing clinical trials.

### 3.3. Glutamine-Mimetics

In analogy with the incorporation of antimetabolites in DNA and RNA leading to cell death, glutamine-mimetics can interfere with the synthesis of purines and pyrimidines. The glutamine analogue 6-diazo-5-oxo-ʟ-norleucine (ʟ-DON) is not only a potent antimetabolite, but it also acts as an inhibitor of glutaminase. In spite of promising preclinical data of efficacy, clinical trials with ʟ-DON revealed excessive toxicities (neurotoxicity, myelosuppression, nausea and vomiting) that did not allow for clinical use [84]. To enhance the effectiveness of ʟ-DON, the glutamine analogue was co-administrated with PEGylated GLS [85]. As expected, the addition of glutamine depletion by the action of GLS resulted in the administration of lower doses of ʟ-DON, leading to an improved toxicity profile. Despite the conclusion of the investigators that this therapeutic approach warrants further evaluation, more recent clinical trials seem to focus more on GLS and GLUD inhibitors.

### 3.4. GLS Inhibitors

More selective inhibition of the glutamine metabolism by GLS and GLUD inhibitors, i.e., without the disruption of the other aspects of the glutamine metabolism, may result in milder side effects, as with ʟ-DON. The GLS-catalyzed conversion of glutamine into glutamate is an early step in the entry of glutamine intermediates into the TCA cycle, and thus, it is critical for the mitochondrial metabolism of both transformed and non-transformed cells. Targeting glutaminase is of special interest as there are two isoforms of the glutaminase gene, i.e., a kidney-type GLS1 and a liver-type GLS2, thereby limiting its side-effects in normal cells. Targeting upregulated GLS1 is a potential strategy in lung cancer as the isoform is frequently upregulated [33].

During the past decade, several inhibitors of GLS, such as compound 968 and BPTES, have been discovered. However, their inhibitory potential has only been described in cancer cells in culture, as their hydrophobic nature poses some challenges to the physiological delivery of the molecules [31]. Another selective GLS inhibitor, is actually under clinical investigation in NSCLC patients (CB-839, phase 1/2 trials NCT02771626 and NCT02071862).

Inhibition of GLS not only limits the mitochondrial metabolism, but it also results in lower concentrations of glutamate and ammonia. Hypothetically, the lower concentration of glutamate and ammonia may potentiate the antitumor effect of glutaminase inhibition. Glutamine-derived glutamate is not only a component of the tripeptide glutathione, but is also required for the uptake of cystine through the SLCA11 antiporter. The reduced concentrations of glutamate and cysteine may result in lower glutathione concentrations in cancer cells. As mentioned before, glutathione protects cancer cells from the oxidative stress associated with their rapid metabolism, and it is thereby critical for survival [90]. As higher GSH levels have been associated with increased resistance to chemotherapy and radiotherapy, there might be a rationale to combine these treatments with a GLS1 inhibitor such as CB-839. Another mechanism by which GLS1 inhibitors may have an impact on the survival of cancer cells is the blockade of ammonia-driven autophagy. Despite the fact that glutamine inhibitors may reduce the replenishment of the TCA cycle intermediates, cancer cells may develop “resistance” by the upregulation of pyruvate carboxylase (PC). As demonstrated by Sellers et al., PC is critical for the proliferation of lung cancers, and overexpression of PC may replenish the TCA cycle, and by this, overcome the GLS1 inhibition, thereby limiting its efficacy in lung cancer cells [43]. A potential strategy for overcoming this resistance is the inhibition of the mitochondrial pyruvate carrier (MPC). Pyruvate, the end-product of upregulated glycolysis in cancer cells, is either reduced in the cytosol by lactate dehydrogenase or imported into the mitochondrial matrix by the mitochondrial pyruvate carrier (MPC) [91]. Metabolic flux analysis, applied to cancer cells after inhibition of the MPC reveal the metabolic flexibility of cancer cells, i.e., their maintenance of oxygen consumption and TCA cycle metabolism. ^13^C experiments revealed that oxidative flux was achieved through an enhanced dependence on glutaminolysis [87]. Thus, pharmacological inhibition of the MPC by small molecule inhibitors such as UK5099 may act synergistically with GLS inhibitors and other treatments targeting glutamine metabolism.

### 3.5. GLUD Inhibitors

Another upregulated and thus potential targetable isoenzyme in lung cancer is GLUD1 [92]. GLUD1 is important for redox homeostasis, as it controls the intracellular concentrations of α-KG, and subsequently fumarate, which functions as an activator of glutathione peroxidase [92]. The inhibition of GLUD1 decreases fumarate levels, which leads to decreased glutathione peroxidase activity, resulting in reduced scavenging of ROS, thereby attenuating cancer proliferation [92]. Investigations of epigallocatechin-3-gallate, the major catechin in green tea, revealed that the compound has anti-tumor effects in lung cancer [86]. The role of epigallocatechin-3-gallate as a maintenance therapy for SCLC is currently under investigation (NCT01317953).

## 4. Conclusions

Understanding the biochemical differences between normal and lung cancer cells is essential for the development of new drugs with therapeutic selectivity. In recent years, progress has been made in understanding glutamine metabolism in cancer. Glutamine has critical roles in the metabolism and behavior of lung cancer cells. Besides the replenishment of the TCA cycle intermediates, glutamine is essential in redox homeostasis, and as a precursor of building blocks such as purines, pyrimidines, amino acids, and lipids. Preferential killing of cancer cells without significant toxicity to normal cells is of utmost importance. Although the direct targeting of glutamine addiction is still in its infancy, isoforms of enzymes that are required for the glutamine metabolism in cancer cells, and that are not commonly expressed in normal cells, are currently under investigation.

## Figures and Tables

**Figure 1 ijms-20-00252-f001:**
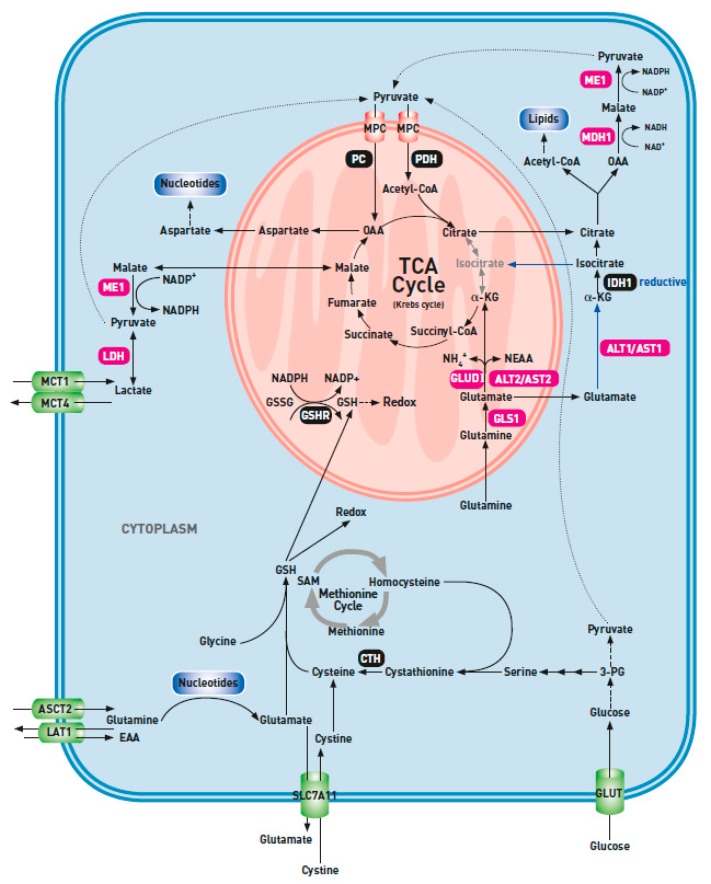
Glutamine metabolism in cancer cells. ALT, alanine aminotransferase; ASCT2, alanine-serine-cysteine-transporter-2; AST, aspartate aminotransferase; CTH, cystathionine gamma-lyase; EAA, essential amino acids; GLS, glutaminase; GLUD, glutamate dehydrogenase; GLUT, glucose transporter; GSH, reduced glutathione; GSHR, glutathione reductase; GSSG, oxidized glutathione; IDH, isocitrate dehydrogenase; α-KG, α-ketoglutarate; LAT1, ʟ-type amino acid transporter; LDH, lactate dehydrogenase; MCT, monocarboxylate transporter; MDH, malate dehydrogenase; ME, malic enzyme; MPC, mitochondrial pyruvate carrier; NADPH, reduced nicotinamide adenine dinucleotide phosphate; NH_4_^+^, free ammonia; OAA, oxaloacetate; PC, pyruvate carboxylase; PDH, pyruvate dehydrogenase; PG, phosphoglycerate; SAM, *S*-adenosylmethionine; SLC7A11, solute carrier family member 7A11 (xCT). Glutaminolysis in pink.

**Figure 2 ijms-20-00252-f002:**
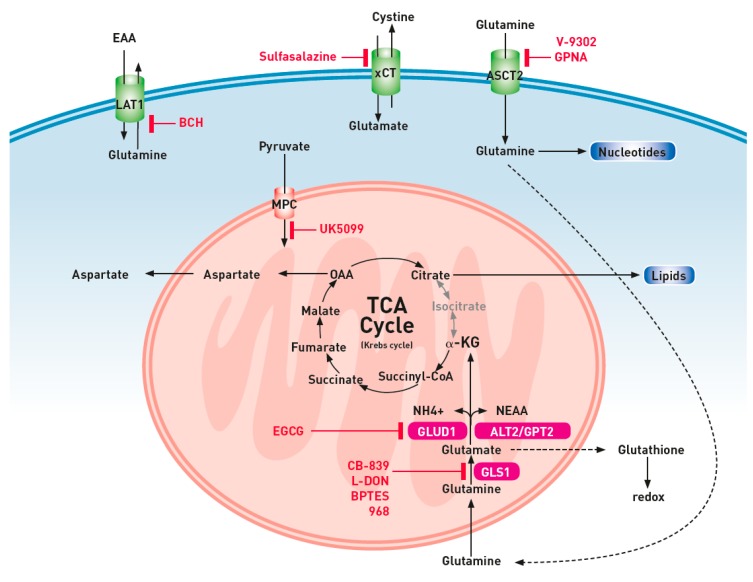
Targeting glutamine metabolism. ALT, alanine aminotransferase; ASCT2, alanine-serine-cysteine-transporter-2; AST, aspartate aminotransferase; BCH, 2-aminobicyclo-(2,2,1)-heptane-2-carboxylic acid; BPTES, Bis-2-(5-phenylacetamido-1,3,4-thiadiazol-2-yl)ethyl sulfide; EAA, essential amino acids; EGCG, epigallocatechin-3-gallate; GLS, glutaminase; GLUD, glutamate dehydrogenase; GPNA, ʟ-glutamyl-p-nitroanilide; α-KG, α-ketoglutarate; LAT1, ʟ-type amino acid transporter; L-DON, 6-diazo-5-oxo-ʟ-norleucine; MPC, mitochondrial pyruvate carrier; OAA, oxaloacetate; SLC7A11, solute carrier family member 7A11 (xCT).

**Table 1 ijms-20-00252-t001:** Strategies for targeting glutamine metabolism.

Target	Mechanism	Drug/enzym	Remarks	Reference
**Glutamine**	Glutamine degradation	Glutaminase	Animal experiments No clinical development (fatal adverse events within 10 days of administration)	[80]
**Glutamine**	Glutamine degradation	Asparaginase	Key component of therapeutic regimens in acute lymphoblastic leukemia Not tested in lung cancer	[81]
**ASCT2** **(SCL1A5)**	Inhibition of glutamine transport	GPNA	Animal experiments and lung cancer cell lines	[82]
		V-9302	Animal experiments and lung cancer cell lines	[83]
**LAT1** **(SCL7A5)**	Inhibition of glutamine-EAA transport	BCH	Lung cancer cell lines	[28]
**xCT** **(SCL7A11)**	Inhibition cystine-glutamate transport	Sulfasalazine	Animal experiments and lung cancer cell lines	[29]
**Glutamine utilizing enzymes**	Glutamine analogue	ʟ-DON	Excessive toxicity, not further developed	[84]
**Glutamine utilizing enzymes**	Glutamine analogue and inhibitor glutaminase	ʟ-DON + PEGylated-glutaminase	Phase II, not further developed	[85]
**GLS1**	Allosteric inhibitor glutaminase1	968 BPTES	Lung cancer cell lines	[31]
	Inhibition of TCA anaplerosis	CB-839	NCT02771626 NCT02071862	
**GLUD1**	Glutathione peroxidase	EGCG	NCT01317953	[86]
**MPC**	Inhibition of pyruvate transport	UK5099	Cell lines	[87]

ASCT2, alanine-serine-cysteine-transporter-2; BCH, 2-aminobicyclo-(2,2,1)-heptane-2-carboxylic acid; BPTES, Bis-2-(5-phenylacetamido-1,3,4-thiadiazol-2-yl)ethyl sulfide; EGCG, epigallocatechin-3-gallate; GLS1, glutaminase; GLUD1, glutamate dehydrogenase; GPNA, ʟ-glutamyl-p-nitroanilide; LAT1, ʟ-type amino acid transporter; ʟ-DON, 6-diazo-5-oxo-ʟ-norleucine; MPC, mitochondrial pyruvate carrier; NCT, identification number for clinical trial; SLC, solute carrier family; TCA, tricarboxylic acid; xCT, cystine transporter.

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
