# Peer review of "Glutamine Addiction and Therapeutic Strategies in Lung Cancer"

_ijms, 2019, doi:10.3390/ijms20020252_

Reviewer 1 Report

Dear,

I have now carefully examined the manuscript entitled “Glutamine addiction and therapeutic strategies in lung cancer” by Vanhove et al. The authors provide a quite exhaustive overview on the different roles of glutamine metabolism in (lung) cancer cells and describe the therapeutic approaches currently developed or evaluated in clinical trials. Overall, the manuscript is well written and the figures are very clear.

Minor points:

Although already referenced, there are some missing citations that must be added. Some notions are also lacking, including the role of glutamine metabolism in stromal cells. 

-to be more exhaustive, the authors should indicate that reductive carboxylation of glutamine can occur under acidosis (in addition to hypoxic and glucose-limiting conditions). See Corbet et al Cancer Res 2014

-indicate MPC in Figure 1 and mention the metabolic rewiring towards glutamine metabolism (supporting a higher sensitivity to glutamine metabolism-related inhibitors). See Yang et al Mol Cell 2014; Vacanti et al Mol Cell 2014

-Edit figure 1 to change “reductief” and “isocitraat” terms

-Besides a role in cancer cells, glutamine metabolism is also important in stromal cells, such as endothelial cells, cancer-associated fibroblasts, immune cell, etc. The authors should, at minimum, mention these points or even better, dedicate a special section to this topic. See Zhang et al Cancer Res 2018 : GFPT2-expressing CAFs mediate metabolic reprogramming in human lung adenocarcinoma; Yang et al Cell Metab 2016 : Targeting stromal GS in tumors…

Author Response

A point-by-point response letter must accompany your revised manuscript. This letter must provide a detailed response to each reviewer/editorial point raised, describing exactly what amendments have been made to the manuscript text and where these can be viewed (e.g. Methods section, line 12, page 5). Please also ensure that all changes to the manuscript are indicated in the text by highlighting or using track changes. If you disagree with any comments raised, please provide a detailed rebuttal to help explain and justify your decision.December 29, 2018

RE: Manuscript ID: ijms-419322

 Dear editor,  

Dear reviewer

 Thank you very much for reviewing our manuscript. We greatly appreciate the reviewers for their constructive comments and suggestions. We have carried out the adaptations that the reviewers suggested and revised the manuscript accordingly. 

 Please find attached a point-by-point response to reviewer’s concerns. We hope that you find the responses satisfactory and that the manuscript is now acceptable for publication.

 Sincerely,

Peter Adriaensens, PhD

Full professor Applied and Analytical Chemistry

Institute for Materials Research

Hasselt University, Belgium

Email : [email protected]

 We strongly appreciate the useful and constructive reviewer’s comments.  Please find below our point-by-point responses:

 Reviewer 1

 to be more exhaustive, the authors should indicate that reductive carboxylation of glutamine can occur under acidosis (in addition to hypoxic and glucose-limiting conditions). See Corbet et al Cancer Res 2014

 In addition to glucose deprivation and hypoxia, Corbet et al. documented that proliferating cancer cells, chronically exposed to acidic conditions, increased the glutamine metabolism to sustain lipogenesis through IDH1 dependent reductive carboxylation of α-ketoglutarate concomitant with fatty acid catabolism (ref 50 and 51).

indicate MPC in Figure 1 and mention the metabolic rewiring towards glutamine metabolism (supporting a higher sensitivity to glutamine metabolism-related inhibitors). See Yang et al Mol Cell 2014; Vacanti et al Mol Cell 2014

 We indicated the MPC in Figure 1 and Figure 2. We added the rationale of inhibition of the mitochondrial pyruvate carrier in subsection 3.4 (Therapeutic perspective of glutamine metabolism, GLS inhibitors) (ref 73 and 74)

 Edit figure 1 to change “reductief” and “isocitraat” terms

We changed the Dutch terms “reductief” and “isocitraat” into English terms “reductive” and “isocitrate” in figure 1

 Besides a role in cancer cells, glutamine metabolism is also important in stromal cells, such as endothelial cells, cancer-associated fibroblasts, immune cell, etc. The authors should, at minimum, mention these points or even better, dedicate a special section to this topic. See Zhang et al Cancer Res 2018 : GFPT2-expressing CAFs mediate metabolic reprogramming in human lung adenocarcinoma; Yang et al Cell Metab 2016 : Targeting stromal GS in tumors.

 The role of the micro-environment is briefly described in section 2.1 Glutaminolysis

 “Indeed, emerging evidence indicates that autophagy in stromal cells such as endothelial cells and cancer associated fibroblasts provides cancer cells with an abundant source of glutamine (ref 38) Emerging evidence indicates that autophagy in stromal cells such as endothelial cells and cancer-associated fibroblasts provides cancer cells with an abundant source of glutamine. A difference in glutamine dependency for K-ras driven lung cancer cells when studied in vitro and in vivo was demonstrated by Davidson et al, supporting the evidence of the complex interaction between tumor and stromal cells on the process of tumour proliferation  (ref 39). More specifically, certain types of stromal cells like cancer-associated fibroblasts and immune cells such as tumor-associated macrophages surrounding the tumor cells might play a role in a higher production of glutamine  when tissue levels are low compared to normal fibroblasts (ref 40)”

Reviewer 2 Report

This is a well written and timely review of glutamine addiction in cancers as it applies to lung cancer.

 Specific comments

 1.       The authors correctly mention a major route by which glutamine is converted to a-ketoglutarate: Namely, conversion of glutamine to glutamate catalyzed by kidney or liver type glutaminases, followed by conversion of glutamate to a-ketoglutarate, either by the glutamate dehydrogenase reaction or by transamination of glutamate.  However, there is another pathway by which glutamine can be converted to a-ketoglutarate.  This pathway (referred to as the glutaminase II pathway) is often completely overlooked, as is the case here. In this alternative pathway, glutamine is transaminated by a kidney or liver type glutamine transaminase to a-ketoglutaramate (KGM). KGM is then converted to a-ketoglutarate by a deamidase known as omega-amidase.  This pathway, should at least be mentioned in the current review.  For a recent review see Cooper et al. Amino Acids 48, 1-20, 2016.

2.       Line 29. Grammatically there is no uptake of the glycolyic pathway.  Change the sentence to read “The increased uptake of glucose and the increased activity of the glycolytic pathway….”

3.       Why are all enzyme names italicized?

4.       P. 4. Line 118.  Names for enzymes need to be standardized. Except in the clinical literature GOT (glutamate oxaloacetate aminotransferase) is rarely used. The most common designation id aspartate aminotransferase. GOT2 is presumably the mitochondrial isozyme, whereas GOT1 is presumably the cytosolic enzyme, but this distinction is not clear and the number is not always given in other parts of the text.  GPT is also more commonly used in the clinical setting. However, the more common name is alanine aminotransferase. Be consistent in your nomenclature.

5.       P. 4. Line 119; 351. Glutamate dehydrogenase is not an isozyme per se, although in humans and higher primates there are indeed two isozymes of this enzyme.

6.        Line 181. TCA cycle.

7.       P. 7. Line 237.  I assume that the enzyme is cystathionine gamma-lyase. Why is this reaction an additional source of cysteine?  Additional to what?

8.       Line 289. It is not glutamine that is being imaged, but fluoroglutamine and its metabolites.

9.       Line 319…..clinical trials seem to focus….

Author Response

A point-by-point response letter must accompany your revised manuscript. This letter must provide a detailed response to each reviewer/editorial point raised, describing exactly what amendments have been made to the manuscript text and where these can be viewed (e.g. Methods section, line 12, page 5). Please also ensure that all changes to the manuscript are indicated in the text by highlighting or using track changes. If you disagree with any comments raised, please provide a detailed rebuttal to help explain and justify your decision.December 29, 2018

RE: Manuscript ID: ijms-419322

 Dear editor,  

Dear reviewer

 Thank you very much for reviewing our manuscript. We greatly appreciate the reviewers for their constructive comments and suggestions. We have carried out the adaptations that the reviewers suggested and revised the manuscript accordingly. 

 Please find attached a point-by-point response to reviewer’s concerns. We hope that you find the responses satisfactory and that the manuscript is now acceptable for publication.

 Sincerely,

Peter Adriaensens, PhD

Full professor Applied and Analytical Chemistry

Institute for Materials Research

Hasselt University, Belgium

Email : [email protected]

 Reviewer 2

The authors correctly mention a major route by which glutamine is converted to a-ketoglutarate: Namely, conversion of glutamine to glutamate catalyzed by kidney or liver type glutaminases, followed by conversion of glutamate to a-ketoglutarate, either by the glutamate dehydrogenase reaction or by transamination of glutamate.  However, there is another pathway by which glutamine can be converted to a-ketoglutarate.  This pathway (referred to as the glutaminase II pathway) is often completely overlooked, as is the case here. In this alternative pathway, glutamine is transaminated by a kidney or liver type glutamine transaminase to a-ketoglutaramate (KGM). KGM is then converted to a-ketoglutarate by a deamidase known as omega-amidase.  This pathway, should at least be mentioned in the current review.  For a recent review see Cooper et al. Amino Acids 48, 1-20, 2016.

 We mentioned the glutamine II pathway in section 2, subsection 2.1 Glutaminolysis although this pathway seems not related with lung cancer.

Line 29. Grammatically there is no uptake of the glycolyic pathway.  Change the sentence to read “The increased uptake of glucose and the increased activity of the glycolytic pathway….”

 We changed the sentence “the increased uptake of glucose and glycolytic pathway have been extensively described” in “ the increased uptake of glucose and the increased activity of the glycolytic pathway have been extensively described”

Why are all enzyme names italicized?

Enzymes were italicized to mark them as enzymes. We changed the italic font into normal font style.

P. 4. Line 118.  Names for enzymes need to be standardized. Except in the clinical literature GOT (glutamate oxaloacetate aminotransferase) is rarely used. The most common designation id aspartate aminotransferase. GOT2 is presumably the mitochondrial isozyme, whereas GOT1 is presumably the cytosolic enzyme, but this distinction is not clear and the number is not always given in other parts of the text.  GPT is also more commonly used in the clinical setting. However, the more common name is alanine aminotransferase. Be consistent in your nomenclature.

We changed the term glutamate oxaloacetate aminotransferase (GOT) into aspartate aminotransferase (AST). We changed the term glutamate oxaloacetate aminotransferase (GPT) into alanine aminotransferase (ALT). Where needed in the text, we specified the cytosolic and mitochondrial form.

P. 4. Line 119; 351. Glutamate dehydrogenase is not an isozyme per se, although in humans and higher primates there are indeed two isozymes of this enzyme.

We removed the term isoenzyme.

Line 181. TCA cycle.

We added the term cycle in this sentence.

P. 7. Line 237.  I assume that the enzyme is cystathionine gamma-lyase. Why is this reaction an additional source of cysteine?  Additional to what?

Indeed, the correct name of the enzyme is cystathionine gamma-lyase and we changed this in the text. This reaction is an additional source of cysteine as the xCT antiporter couples the export of glutamate and the import of cystine. Once in the cell, cystine is reduced into cysteine explaining the term additional source.

Line 289. It is not glutamine that is being imaged, but fluoroglutamine and its metabolites.

Indeed, we changed the previous sentence into : In this context, pretreatment evaluation of glutamine uptake by ASCT2, using positron emission tomography with labelled “glutamine-analogues”, may select patients that benefit from GPNA treatment.

Line 319…..clinical trials seem to focus….

We corrected the spelling error